# Integrating Microarray Data and Single-Cell RNA-Seq Reveals Key Gene Involved in Spermatogonia Stem Cell Aging

**DOI:** 10.3390/ijms252111653

**Published:** 2024-10-30

**Authors:** Danial Hashemi Karoii, Hossein Azizi, Thomas Skutella

**Affiliations:** 1Department of Cell and Molecular Biology, School of Biology, College of Science, University of Tehran, Tehran 14174-66191, Iran; d.hashemi.karoii@ut.ac.ir; 2Faculty of Biotechnology, Amol University of Special Modern Technologies, Amol 46156-64616, Iran; 3Institute for Anatomy and Cell Biology, Medical Faculty, University of Heidelberg, Im Neuenheimer Feld 307, 69120 Heidelberg, Germany; thomas.skutella@uni-heidelberg.de

**Keywords:** spermatogonia stem cell, microarray, bioinformatics, gene ontology, germ cell

## Abstract

The in vitro generation of spermatogonial stem cells (SSCs) from embryonic stem cells (ESCs) offers a viable approach for addressing male infertility. A multitude of molecules participate in this intricate process, which requires additional elucidation. Despite the decline in SSCs in aged testes, SSCs are deemed immortal since they can multiply for three years with repeated transplantation. Nonetheless, the examination of aging is challenging due to the limited quantity and absence of precise indicators. Using a microarray, we assessed genome-wide transcripts (about 55,000 transcripts) of fibroblasts and SSCs. The WGCNA approach was then used to look for SSC-specific transcription factors (TFs) and hub SSC-specific genes based on ATAC-seq, DNase-seq, RNA-seq, and microarray data from the GEO databases as well as gene expression data (RNA-seq and microarray data). The microarray analysis of three human cases with different SSCs revealed that 6 genes were upregulated, and the expression of 23 genes was downregulated compared to the normal case in relation to aging genes. To reach these results, online assessments of Enrich Shiny GO, STRING, and Cytoscape were used to forecast the molecular and functional connections of proteins before identifying the master routes. The biological process and molecular function keywords of cell–matrix adhesion, telomerase activity, and telomere cap complex were shown to be significantly altered in upregulated differentially expressed genes (DEGs) by the functional enrichment analysis. According to our preliminary research, cell-specific TFs and TF-mediated GRNs are involved in the creation of SSCs. In order to maximize the induction efficiency of ESC differentiation into SSCs in vitro, hub SSC-specific genes and important SSC-specific TFs were identified, and sophisticated network regulation was proposed. According to our research, these genes and the hub proteins that they interact with may be able to shine a light on the pathophysiologies of infertility and aberrant germ cells.

## 1. Introduction

Infertility is a significant global health issue, affecting an estimated 10% of couples worldwide, with male factors being a main or contributing cause in over 50% of cases [1,2]. Male infertility is a multifactorial clinical disease characterized by complicated genetic variables, with azoospermia being the most prevalent hereditary contributor to male infertility [3,4]. Spermatogonial stem cells (SSCs) are the progenitor cells of sperm and serve as the foundation for spermatogenesis and male fertility. Consequently, SSCs are seen as a viable option for the restoration of compromised or injured spermatogenesis, and SSC transplantation is a potential method for the treatment of male infertility [4,5,6,7]. Nonetheless, the quantity of SSCs is very limited, and the long-term culture and proliferation of SSCs remain unavailable. Currently, research in both mice and humans indicates that embryonic stem cells (ESCs) have the capacity to generate putative primordial germ cells (PGCs), which may subsequently transform into SSCs [8,9]. Nonetheless, these results either include a convoluted induction process with ambiguous induction variables or exhibit a poor induction efficiency, while the restoration of SSC formation in vitro continues to pose a significant problem.

Stem cells are found in almost all adult tissues, often situated in the basement membrane, where they engage with adjacent stromal cells which create a safeguarded area known as the stem cell niche [10,11]. Deviant control of the stem cell/niche compartment may significantly impact tissue maintenance, perhaps resulting in manifestations of aging. A key inquiry is whether aging arises from the decline in stem cells’ self-renewal capacity, the regulatory role of the niche, or a combination of both factors [12]. A fundamental prerequisite for a definitive assessment is a functional experiment whereby stem cells are extracted from their respective young or old niches, then transplanted into a heterologous recipient environment, and achieve total replacement of the dependent tissue or system. This criterion now applies just to hematopoiesis, epidermis/hair, and spermatogenesis [13,14].

The perpetual multiplication of stem cells invites speculation over their potential immortality. One idea suggests that stem cells possess a limited replicative capacity that is depleted with aging [15]. When hematopoietic stem cells (HSCs) are serially transplanted into lethally irradiated recipients, their regenerative capacity is depleted by the fifth passage, hematopoiesis fails to return to a normal state, and the stem cell population does not recover to the levels observed in unmanipulated animals [16]. As a result, the possible immortality of hematopoietic stem cells, the most well studied adult stem cells, has been scrutinized. Furthermore, investigations of the stem cell/niche unit and clonal development in the hematopoietic system are intricate due to the positioning of the HSC inside the medullary cavity [17,18].

Spermatogenesis is a self-renewing process sustained by SSCs and demonstrates age-related declines that eventually lead to infertility [18]. The SSC/niche compartment in the testis is easily accessible. The SSC transplantation approach facilitates the clonal assessment of stem cell number and quality [19,20,21,22,23,24]. Consequently, we used this approach to evaluate the notion that deficiencies in SSC quantity or functionality induce age-related infertility in mice. Finally, many research efforts have examined this microarray in human male germ cells to identify aging-related genes. This experimental work investigated the expression of microarray genes in SSCs, using bioinformatic tools like gene ontology, protein–protein interaction, signaling pathways, and single-cell RNA sequencing.

## 2. Results

### 2.1. Human Spermatogonia Selection and Culture

Spermatogonia were extracted and concentrated by CD49f-MACS and matrix selection (particularly collagen nonbinding/laminin binding) from orchiectomies conducted to acquire patient data relevant to SSC cultures. In the selected cell populations, spermatogonia were likely the predominant cells, shown by positive DDX4 (VASA) and negative VIMENTIN immunocytochemistry in the initial cultures. In long-term cultivated cells, VASA staining was diminished, and they had found VIMENTIN positivity. The morphology of pure spermatogonia, derived from many individuals, exhibited consistency irrespective of the patients’ age or the culture time. The fundamental rationale for this was the spherical form and size of around 6–12 μm in diameter, together with the elevated nucleus-to-cytoplasm ratio. This ratio may be identified by a prominent, luminous cytoplasmic ring situated between the spherical nucleus and the external cell membrane. All cell cultures exhibited interconnected pairs, chains, and tiny clusters of spermatogonia, joined by intercellular bridges. Various types of cells were found inside the cultures, including bigger cells reaching 12–14 μm in diameter. The cells exhibited an oval morphology and a decreased nucleus-to-cytoplasm ratio. The unselected cell population exhibited a substantial reduction in htFibs. The htFibs, demonstrating substantial growth in primary cell cultures, were successfully separated from the nonselected cell fractions. Figure 1 illustrates examples of spermatogonial cultures devoid of htFibs.

### 2.2. Microarray Analysis of Gene Expression in SSC Versus Fibroblast

A microarray was used to analyze around 1100 G-protein transcripts. Utilizing microarray analysis, we identified 12 genes that were upregulated and 18 genes that were downregulated in three particular kinds of stem cells (SSCs) and fibroblasts. This information is shown in the figure. The microarray analysis of three SSC human samples indicated that the genes *TNFRSF10B*, *ATF2*, *FAS*, *MAP4K5*, *ATF7*, *TNFRSF10D*, *IGF2R*, *PRKCA*, *AIFM1*, *AKT2*, *HSPA8*, *BAX*, *MAPK14*, *MAP4K4*, *JUN*, *XIAP*, *MAP2K3*, *FOS*, *TNFRSF1A*, *BAG3*, *MCL1*, *FADD*, and *CFLAR* exhibited downregulation, while *PRKCE*, *HSPA2*, *BAG1*, *BCL2L11*, *CREM*, and *HSPA1L* demonstrated upregulation (Figure 2 and Table 1).

### 2.3. Functional Enrichment Analysis of DEGs

The five most significant terms from the gene ontology enrichment analysis indicated that, within the biological process category, the upregulated and downregulated DEGs were associated with GO:0032202 telomere assembly, GO:0070200 establishment of protein localization to telomeres, GO:0032211 negative regulation of telomere maintenance via telomerase, GO:0070198 protein localization to chromosome telomeric regions, and GO:2000279 negative regulation of DNA biosynthetic processes. In the MF category, the upregulated and downregulated DEGs were associated with GO:0071209 U7 snRNA binding, GO:0010521 telomerase inhibitor activity, GO:0030156 benzodiazepine receptor binding, GO:0098639 collagen binding linked to cell–matrix adhesion, and GO:0003720 telomerase activity. In the CC category, the differentially expressed genes (DEGs) that were upregulated or downregulated were enriched in the following gene ontology (GO) terms: GO:1990879 CST complex, GO:0031379 RNA-directed RNA polymerase complex, GO:0000782 telomere cap complex, GO:0000783 nuclear telomere cap complex, and GO:0140445 chromosomal telomeric repeat region (Figure 3).

### 2.4. PPI Network 

A PPI network was constructed for the differentially expressed genes (DEGs) in spermatogonial stem cells (SSCs) using data from the STRING database. This analysis positioned a total of 168 genes within the PPI network. The PPI network had 142 nodes and 450 edges, exhibiting a PPI enrichment *p*-value of less than 0.01. A network of 30 DEGs and their neighboring genes was constructed via FunRich. Subsequently, the MCODE plugin was used to identify the significant modules. The first four functional clusters of modules were selected according to their MCODE scores: module 1 (MCODE score = 21.826), module 2 (MCODE score = 3.333), module 3 (MCODE score = 3), and module 4 (MCODE score = 2.8. DAVID was used to perform a KEGG pathway analysis for each module. The KEGG pathway analysis of module 1 indicated the participation of these genes in cell cycle regulation, DNA replication, and oocyte meiosis. Module 2 consisted of four nodes and five edges, with genes showing enrichment in ribosomes. Module 3 had three nodes and three edges, including genes linked to amino acid synthesis, the HIF-1 signaling pathway, and carbon metabolism. Module 4 had six nodes and seven edges, including genes associated with fat digestion and absorption, along with the PPAR signaling pathway. The PPI enrichment *p*-value for each module was below 0.05.

The cytoHubba plugin identified twenty-one hub genes associated with ovarian cancer, namely TEN1, FN1, LSM11, SMCO4, ACD, PLEKHS1, TERT, CYREN, CLPTM1L, SEMG2, MIA, ITGAD, FSD1L, FNDC4, LACRT, CTC1, RIMBP3B, MKRN1, ITGA10, SFTPC, and WRAP53, based on their elevated degree scores, the STRING online database was used to construct the PPI network of the key genes. Additionally, FunRich software (version 3.1.4) was employed to construct the interaction network for the core genes and their associated genes (Figure 3B). The PPI network of the hub genes comprised 10 nodes and 45 edges. The average local clustering coefficient of the network was 1, indicating a high degree of clustering. The PPI enrichment *p*-value was determined to be below 0.01, indicating a substantial enrichment of protein interactions. Furthermore, the results of the gene co-expression analysis of the 10 hub genes suggest that these genes likely participate in active interactions with each other (Figure 4).

### 2.5. Construction of Weighted Gene Co-Expression Modules

We constructed gene co-expression networks using the WGCNA algorithm to identify functional clusters in SSCs. A recent investigation identified five modules in the SSC, each assigned a distinct hue. It is essential to observe that a single module, which was excluded from any cluster, was denoted by the color gray. Subsequently, we produced a heatmap to evaluate the association between modules and attributes. Figure 2B and Figure 3B illustrate the links between modules and traits. This research indicates that the brown module in the SSC has the most robust connection with normal tissues (brown module: r = 0.58, *p* = 9 × 10^−51^; pink module: r = 0.8, *p* = 1 × 10^−10^) (Figure 5).

### 2.6. Testicular Cell Composition during Testis Development in Humans

To characterize the cellular diversity of the testis during human development, we conducted single-cell RNA sequencing on testicular cells from males aged 1, 2, and 7 years, integrating these data with publicly available databases. We constructed a developmental timeline for testis maturation and conducted a comparative analysis across different age groups. The datasets spanned various developmental stages, including prenatal (embryonic weeks 6–16), neonatal (postnatal days 2 and 7), prepubertal, and peri- to postpubertal ages (11, 13, 14, 17, and 25 years). A total of 82,220 individual testicular cells were preserved for further examination.

Using recognized testicular cell markers, 17 unique cell clusters were discovered from the UMAP embedding of the pooled datasets (Figure 1). Germ cells (DDX4+, ID4+), Sertoli cells (AMH+, SOX9+), myoid cells (ACTA2+, RGS5+), macrophages (CD14+), and endothelial cells (PECAM1+) were among the cellular components, as shown in Figure 1. Clusters 13, 14, and 15 showed co-expression of DLK1 and NR2F2, which indicated Leydig cells and somatic precursors (Figure 1). The somatic and germ cell composition varied with age, with somatic cells (Sertoli, Leydig, and somatic precursors) predominating during the prenatal (W6–W16), neonatal (2D–7D), and prepubertal stages (1Y–7Ys). During puberty, the number of germ cell stages increased (clusters 2–4) (Figure 1). These shifts in cell composition align with the anatomical and functional changes that germline and somatic cells undergo throughout testis development.

We re-clustered 8,140 germ cells from clusters 1 through 4 in order to better under-stand the diversity of germ cells throughout testis development. Fifteen cell clusters representing various phases of spermatogenesis were found via unbiased clustering using UMAP (Figure 1). In clusters 1 through 5, undifferentiated spermatogonia (Undiff SPG) or SSC/progenitors were identified by the expression of UTF1, ID4, and NANOS3, among other markers. Differentiating spermatogonia (Diff SPG), represented by cluster 6, lacked meiotic gene expression but had CKIT and STRA8 expression. SYCP3 and SPO11 were overexpressed in clusters 7 and 8, which correspond to spermatocytes (SCytes). TNP1 and PRM2 were expressed by clusters 9 to 15, which included round and elongated spermatids (SPtids) (Figure 6). This suggests that spermatogenesis had finished by the age of 14.

### 2.7. Analysis of Single-Cell RNA Sequencing Indicates That the Development of Human Spermatogonia Is Marked by Aging Signaling Pathways

To ascertain functional alterations during SSC development, we consolidated the five clusters indicative of SSC/progenitors into a single cluster and conducted pairwise DEG analysis across age groups, subsequently performing pathway enrichment analysis with GSEA. The enriched pathways were found using a stringent false discovery rate threshold of less than 0.05. Figure 2A demonstrates a notable enrichment of the GSEA signature gene set related to OXPHOS and c-myc targets throughout the prenatal stages (embryonic weeks 6 and 16 (W6_W16)) in contrast to the neonatal stages. The abundance of OXPHOS-associated genes aligns with the metabolic demands crucial for the specification of PGCs from pluripotent stem cells in mice and corresponds with prior findings from human and pig PGCs. We detected the preservation of OXPHOS-associated gene enrichment when comparing D2_D7 (neonatal stage) to one year of age.

### 2.8. Scoring Cell–Cell Interactions Using Known Ligand–Receptor Interactions

Following the identification of several cell types, we proceeded to assess potential interconnections among all cell types present in the tumor microenvironment. We used a collection of around 1800 validated and scientifically substantiated interactions. These interactions include receptor–ligand interactions from many families, including chemokines, cytokines, receptor tyrosine kinases (RTKs), tumor necrosis factors (TNFs), and extracellular matrix (ECM)–integrin interactions. Additionally, we included recognized interactions of B7 family members manually, given their importance in cancer immunology.

We examined each of the six syngeneic tumor models to uncover analogous cell–cell interactions by determining instances when both elements of a certain ligand–receptor interaction were expressed by cell types inside the tumor microenvironment (Figure 7). The interactions were scored by computing the product of the average receptor expression and the average ligand expression in the specific cell types being studied (scoring ligand–receptor interactions). To mitigate the risk of false negatives resulting from zero dropouts, we used the average expression value for each cell type. By producing scores for each tumor, we determined the average interaction score across the tumor models. This enabled us to discern conserved connections. After analyzing various cell–cell interactions (around 1500 ligand–receptor pairs translated to mouse homologs and 64 cell type combinations), we evaluated the statistical significance of each interaction score employing a one-sided Wilcoxon rank-sum test and applied the Benjamani–Hochberg multiple hypothesis correction. While we analyzed interactions for all recognized cell types, our focus was only on those in which either cancer-associated fibroblasts (CAFs) or macrophages emitted the ligand. This decision was based on both cell types being the principal source of various ligands. Furthermore, we performed an exhaustive examination of all interactions involving tumor cells.

We identified several extracellular matrix (ECM)-related interactions that positively correlated with tumor progression. A notable correlation was observed between the rate of tumor growth and interactions between cancer-associated fibroblasts (CAFs) and endothelial cells. These interactions promote the synthesis of collagens, which bind to Cd93 or integrin receptors on neoplastic cells. Additionally, a strong association was found between the tumor growth rate and the presence of intercellular adhesion molecules (ICAMs) that interact with integrins, suggesting the potential for further adhesion-related interactions. The expression of ADAM12 and ADAM15 proteases also showed distinct correlations with tumorigenesis. ADAM12, when interacting with its integrin substrates, exhibited an inverse relationship with tumor growth, whereas ADAM15’s interaction with integrin beta 3 (ITGB3) was positively correlated with tumor growth. We also uncovered several chemokine and cytokine interactions related to tumor progression. Elevated expression of CCL11 by tumor cells, which interacted with CCR5 or CXCR3 receptors on both macrophages and tumor cells, was positively associated with tumor growth. Conversely, a negative correlation was observed between the production of interleukin 1 alpha (IL1A) by CAFs and the expression of its receptors—IL1R1, IL1R2, and IL1RAP—on tumor cells. This interaction appears to influence the tumor growth dynamics. Numerous interactions involving receptor tyrosine kinases (RTKs), which were relatively few among the predominant interactions (Figure 7), demonstrated a correlation with the rate of tumor growth. An autocrine relationship occurred among tumor cells, marked by the concurrent secretion of epidermal growth factor (EGF) and the expression of Erbb3 receptors. This link showed a direct correlation with tumorigenesis. Furthermore, a clear correlation existed between the secretion of platelet-derived growth factor (PDGF) by cancer-associated fibroblasts (CAFs) and the presence of vascular endothelial growth factor (VEGF) ligands. These ligands can bind to both PDGF receptors (Pdgfrb) and VEGF receptors (Kdr/Vegfrr2) present on tumor cells. This connection subsequently promotes tumor growth (Figure 8).

In conclusion, we noted that several interactions had a substantial correlation with a certain phenotype. The interactions engaged the same receptor but varied in the ligands used. This discovery prompted an inquiry into whether a particular ligand or receptor is responsible for the association, rather than the physical contact itself. To investigate this hypothesis, we computed Spearman correlations between the expression of the receptor alone or the expression of the ligand alone and the rate of tumorigenesis. Our findings indicate that interaction scores demonstrating a substantial association with the phenotype are often related to either a strong correlation with receptors or a high correlation with ligands. Given that interaction scores are affected by the expression levels of both receptors and ligands, it is reasonable to conclude this outcome, as these two factors are interconnected and not mutually exclusive. Nonetheless, there were specific instances where the interaction score exhibited a significant correlation with the tumor growth rate, despite the absence of a robust correlation between the receptor and the ligand, particularly in the area of the plot where the expression of both ranged from −0.5 to 0.5. Furthermore, we have seen instances where the expression of receptors and ligands exhibited distinct and significant correlations, located in the upper left and lower right quadrants of the picture. This indicates that the relationship seen in the interaction scores is not only due to the correlations between ligands and receptors (Figure 9).

## 3. Discussion

Aside from acting as the reservoir that sustains male fertility, spermatogonial stem cells (SSCs) hold significant therapeutic potential across various species, with applications ranging from clinical fertility restoration to biobanking efforts aimed at protecting endangered species [20,22,24,25,26]. Most of thecell type in mammalian testes suggests that the implementation of SSC-based technologies will largely depend on the efficacy of in vitro growth [27,28]. The exposure of SSCs to an in vitro environment always induces biological alterations, thereby leading to a reduction in their regeneration ability. This study identifies a distinct set of differentially expressed genes in undifferentiated spermatogonia obtained directly from the testes compared to those cultured in vitro for 10 weeks. Our findings reveal significant dysregulation in critical biological processes, such as metabolism and DNA repair, suggesting that prolonged culture adversely affects cellular function. Notably, we observed that the proteasomal degradation of ubiquitinated proteins becomes compromised with an extended culture duration. The insights presented in this brief research report lay an essential groundwork for evaluating the safety of cultured spermatogonial stem cells (SSCs) for therapeutic applications. Furthermore, these results underscore the urgent need to optimize culture conditions to better replicate the natural environment of the testis. Our data reveal differential gene expression between SSCs derived from testes and those cultured; however, it is crucial to emphasize that most SSC signature genes, including Etv5, Gfra1, Lhx1, Bcl6b, and Ret, maintain elevated expression levels in cultured cells at the analyzed 10-week time point. This corresponds with the sustained ability of some cells to regenerate spermatogenesis after transplantation into a recipient testis and illustrates the significance of these primary cultures for investigative studies on SSC function, especially at initial time periods or passage numbers. This contrasts with the immortalized GC-1 “spermatogonia” cell line, which, when compared to scRNAseq data from mouse testes, shows minimal similarity to spermatogonia or spermatocytes, instead exhibiting a gene expression profile which more closely aligns with somatic cells in the testes.

The evaluation of differentially expressed genes between testicular and cultured spermatogonia revealed a disturbance in metabolism-regulating genes. Gene ontology research specifically revealed “oxidative phosphorylation” and “oxidative reduction activities” as pathways that were elevated in undifferentiated spermatogonia derived from cultures [29,30]. This gene group included Cox6a1 and Ndufb7, which are essential components of the mitochondrial electron transport chain necessary for aerobic metabolism [31,32]. Considering that various independent groups have shown that enhancing glycolysis in a spermatogonial culture markedly improves SSC maintenance, it follows that an extended culture duration leads to a transition towards oxidative phosphorylation, which is subsequently linked to the noted decrease in regenerative capacity [33]. This is a theory we have already proposed. Nonetheless, this contradicts a recent work by Kanatsu-Shinohara et al. that observed the long-term “aging” of SSCs in in vitro cultures. This research indicated an elevation in glycolysis with an extended culture duration (comparing cultures sustained for 5 vs. 60 months), which the authors attributed to dysregulated Wnt7b expression, leading to subsequent impacts on Jnk (Mapk8) and Ppargc1a [34]. Predictably, after 60 months of in vitro development, SSCs transplanted into recipient testes were incapable of regenerating spermatogenesis, although exhibiting characteristics of the glycolytic metabolism [35]. Several potential reasons exist for the discrepancies in results between this work and ours, the primary one being that Kanatsu-Shinohara et al. compared spermatogonia at two different time periods in culture rather than contrasting those in culture with those from testes [36]. In their investigation, the “young” cultures were sustained in vitro for 5 months, a duration during which many biological alterations had likely transpired compared to the initial cell population obtained from the testes. Additional distinctions include the age of the animals from whom spermatogonia were extracted for culture establishment (adult vs. postnatal) and the genetic lineage of the mice (C57BL6J versus DBA/2) [37]. Nonetheless, together these findings indicate that extended exposure to an in vitro environment significantly affects SSC metabolism, adversely impacting the long-term regeneration potential [38,39].

The finding of downregulated DNA repair mechanisms in undifferentiated spermatogonia after 10 weeks of in vitro culture is particularly significant when evaluating the safety of using cultured SSCs for therapeutic applications. The expression of the homology-directed repair enzymes Brca2 and Rad51 was dramatically diminished. Consistent with this, Kanatsu-Shinohara et al. observed a considerable rise in DNA damage indicators in undifferentiated spermatogonia with an extended culture duration. Considering the association between DNA damage in spermatozoa and male infertility [40], it is imperative to further explore the relationship between SSC culture and DNA damage, as well as the outcome of “damaged” SSCs post transplantation into the testis.

Our analysis revealed a notable downregulation in the expression of many proteasomal subunits and other elements of the ubiquitin-mediated proteolysis pathway in undifferentiated spermatogonia after 10 weeks of culture. The ramifications of this diminished expression on proteasome activity are evident in the significant buildup of ubiquitinated proteins identified in undifferentiated spermatogonia from culture, in contrast to those obtained straight from testes. While it is generally recognized that balance between protein production and degradation (proteostasis) is essential for male fertility, the specific needs for proteostasis in the SSC population remain largely unexplored. Interestingly, in hematopoietic stem cells, proteostasis is crucial for optimal function, since the buildup of misfolded proteins directly hinders self-renewal ability and stem cell quiescence. This suggests that a comparable mechanism could be present in SSCs and that proteasome failure associated with an extended culture duration might correlate with a decline in self-renewal potential. It is also important to note that oxidative stress may trigger protein unfolding/misfolding and subsequent protein aggregation. Our analyses indicate that a network of genes associated with “oxidative phosphorylation” was upregulated in cultured spermatogonia, suggesting that this, combined with the dysregulated function of the ubiquitin-mediated proteolysis pathway, may create an optimal condition for the accumulation of misfolded proteins and protein aggregates. NOA is a rare condition, and obtaining a large cohort of affected individuals can be difficult. A limited sample size reduces the statistical power of a study and may not capture the full genetic or clinical variability of the condition, potentially leading to the underrepresentation of certain subtypes of NOA or relevant gene mutations. NOA includes several subtypes, such as Sertoli cell-only syndrome, maturation arrest, and hypospermatogenesis. If a study does not stratify patients by these specific subtypes, the results may be skewed or diluted, as each subtype may have distinct underlying genetic or molecular mechanisms.

In conclusion, the extended culture of SSCs unequivocally induces many biochemical alterations that adversely affect the cells’ regenerative ability. Nonetheless, primary cultures of undifferentiated spermatogonia are an essential resource for investigative studies into the molecular control of spermatogonial stem cell activity and provide significant potential as part of therapeutic strategies for fertility restoration or conservation efforts. This publication presents the first comprehensive investigation of gene expression alterations in undifferentiated spermatogonia after a culture period, serving as a vital resource for the future advancement of this approach.

## 4. Materials and Methods

### 4.1. Study of Testicular Tissue and Experimental Design

Following our recent studies, this study was conducted from October 2016 to September 2017, using testicular samples acquired from three adult males [19,41]. The research conducted using human materials at this site received authorization from the local ethics committee of the Committee of the Medical Faculty of the University of Heidelberg, reference number S-376/2023, and the Amol University of Special Modern Technologies Ethics Committee, with reference number Ir.asmt.rec.1403.006. Furthermore, written consent was obtained from all human participants after supplying them with pertinent information. The patients’ ages ranged from 23 to 67 years. The healthy donor tissue included a variety of acquired elements.

This study examined the gene expression profiles of short-term (<2 weeks post matrix selection) SSC cultures and long-term (>2 months, up to 6 months) haGSC cultures derived from the testicular tissues of five guys to elucidate the properties of testicular adult stem cells. Microarray analysis was used to compare these profiles with those of human embryonic stem cells and human fibroblasts.

### 4.2. Selection and Cultivation of haGSCs

Subsequent to the excision of the tunica albuginea, the human testicular tissues were meticulously manipulated to segregate the tubules. Each sample underwent tubule degradation using the following enzymes: 750 U/mL collagenase type IV (Sigma, Darmstadt, Germany), 0.25 mg/mL dispase II (Roche, Basel, Switzerland), and 5 μg/mL DNase. This was performed in HBSS buffer with Ca++ and Mg++ (PAA, Farnborough, UK) for 30 min at 37 °C, with moderate agitation. The objective was to generate a single-cell suspension. The digestion process was then slowed by the introduction of 10% ES cell-qualified fetal bovine serum (FBS, Thermo Fisher Scientific, Bremen, Germany). The most effective cell cultures were subcultured at a frequency of 1:2 every two to three weeks. It was essential to prevent excessive cell dilution and constantly maintain the optimal cell count in the wells.

Inclusion Criteria: Typically, donors could be adult males of reproductive age, as SSCs are relevant in this population. Regarding their general health, they should be healthy males with no underlying medical conditions affecting fertility. To study normal SSC behavior, donors with no history of infertility, testicular cancer, or genetic reproductive issues should be selected. All participants should provide informed consent, agreeing to the use of their biological samples.

Exclusion Criteria: Donors with known genetic, infectious, or systemic diseases that could affect SSCs (such as mumps orchitis, cryptorchidism, or cancer) would be excluded. Any donor currently on medications affecting the reproductive system, like hormonal therapies or chemotherapy, would likely be excluded. Smoking, alcohol abuse, or drug use might also be grounds for exclusion due to their potential impact on sperm quality and SSC function.

### 4.3. Human Fibroblast Cultivation

Human fibroblasts were obtained from the dermis of the scrotum, and a primary cell line was established using DMEM high glucose, 10% FBS Superior (Biochrom, Jena, Germany), 200 μM L-glutamine (PAA, UK), 1% nonessential amino acids (PAA, UK), and 100 mM β-mercaptoethanol (Invitrogen, Thermo Fisher Scientific, Germany).

### 4.4. Isolation of Individual Cells from a Population of hSSCs

Each sample was subjected to a washing procedure using culture media to isolate the spermatogonial cells from the accompanying monolayer of somatic cells or feeder layer on a culture plate. After a gentle resuspension, the cells were deposited onto a tiny culture plate (diameter = 3.5 cm) as a single-cell suspension. The dish cover was placed on the warmed (37 °C) working platform of a Zeiss inverted microscope, alongside the micromanipulation gear. The cells were methodically collected using a micromanipulation pipette at a magnification of 20×. The distinctive structure of spermatogonia cultivated for a brief duration was readily observable. The primary characteristics of these cells were their spherical shape, measuring around 6–12 μm in diameter, and a pronounced nucleus-to-cytoplasm ratio. A conspicuous brilliant cytoplasmic ring situated between the rounded nucleus and the outer cell membrane might facilitate detection.

### 4.5. Collection of Single Cells from Enzymatically Degraded Typical haGSC Colonies

To examine the individual cells inside the haGSC colony, we used enzymes to dissociate a standard haGSC and hESC colony, as well as a highly proliferating hFibs colony, into single cells. We then selected individual cells manually using a micromanipulation approach to examine their gene expression levels at the single-cell level. This method was developed to collect data on the specific cellular attributes of key genes linked to germline and pluripotency. The objective was to analyze the diversity in gene expression among chosen cells from a typical haGSC colony. Furthermore, the objective was to cultivate colonies that demonstrated the most advantageous gene expression patterns associated with germ and pluripotency.

### 4.6. Immunocytochemistry Staining

Cells were cultured and subjected to treatment with 4% paraformaldehyde over 24 plates. Subsequent to PBS washing, samples were permeabilized using 0.1% Triton in PBS and then blocked with 1% BSA in PBS. Following the elimination of the blocking solution, the cells were subjected to treatment with primary antibodies. Subsequent to 30 rinses, the protocol included incubation with species-specific secondary antibodies conjugated to diverse fluorochromes. Cells were counterstained for 5 min at room temperature with 0.2 g/mL DAPI (4′,6-diamidino-2-phenylindole) before fixation with Mowiol 4–88 reagent. The lack of all primary antibodies in the sample functioned as a negative control for all indications. A confocal Zeiss LSM 700 microscope was used to examine the tagged cells, and images were acquired using a Zeiss LSM-TPMT camera (Zeiss LSM 880, Munich, Germany).

### 4.7. Microarray Analysis

Using the RNeasy Mini Kit (Qiagen, Hilden, Germany), RNA was extracted from testicular fibroblasts (hFibs; negative control), long-term haGSC cultures, short-term spermatogonia, and the hESC line H1 (positive control). Subsequently, an amplification procedure was conducted using the MessageAmp aRNA Kit (Ambion, Thermo Fisher Scientific, Germany). A micromanipulation equipment was used to catch 200 cells per probe for each sample. After that, the cells were kept at −80 °C in 10 μL of RNA direct lysis solution. The samples were examined at the University of Tübingen Hospital’s microarray facility in Germany. Affymetrix’s Human U133 + 2.0 Genome oligonucleotide array was used to examine gene expression. MicroDiscovery GmbH in Berlin, Germany, received the raw data (CEL files) for biostatistical analysis and normalization.

### 4.8. Identification of DEGs

GEO2R is a web-based program that facilitates the comparison and analysis of two separate sample sets exposed to similar experimental conditions. This study first examined the selected datasets of ovarian cancer (OC) tissues and normal tissues using GEO2R. Subsequently, the analysis findings were downloaded in Microsoft Excel (Versions 2022) format. Genes meeting the threshold of an adjusted *p*-value below 0.05 and an absolute log fold change above 1.0 were designated as DEGs. The FunRich application (version 3.1.3) was used to graphically depict the overlap of DEGs. The heatmap of the DEGs was produced using ClustVis, an online tool [6,23,24,42,43,44,45,46,47].

### 4.9. Sorting Groups of Proteins

The analysis of differentially expressed genes across the three research groups was performed using the online application ArrayMining. The gene list was analyzed with PANTHER, a methodology for gene ontology analysis.

### 4.10. Gene Ontology Analysis

The discovery of unique cluster markers and DEGs inside the selected clusters was executed using the “FindAllMarkers” and “FindMarkers” functions in Seurat, respectively. The DAVID Bioinformatics Resources (V6.8) [48] platform was used for the qualitative analysis of the gene lists.

### 4.11. Identification of Key Co-Expression Modules Using WGCNA

Co-expression networks provide network-based gene screening methodologies for the discovery of possible hSSC aging pathways. We used the WGCNA [49] program in R to construct gene co-expression networks using the gene expression data profiles of hSSC. The WGCNA approach was used to analyze gene modules that have significant correlations across samples, with the objective of linking these modules to extrinsic characteristics seen in the samples. The function pickSoftThreshold was used to choose soft powers β = 3 and 20 for the construction of a scale-free network. Subsequently, the adjacency matrix was constructed using the formula aij = |Sij|β, where aij means the adjacency matrix between gene i and gene j, Sij signifies the similarity matrix derived from Pearson correlation of all gene pairs, and β indicates the soft power value. The adjacency matrix was then transformed into a topological overlap matrix (TOM) along with its associated dissimilarity (1-TOM). A dendrogram for hierarchical clustering was then generated using 1-TOM. This dendrogram was used to classify gene expressions exhibiting similar patterns into separate gene co-expression modules. To identify functional modules in a co-expression network, the linkages between modules and clinical traits were evaluated, using the technique outlined in a previous study. Consequently, modules exhibiting a high correlation coefficient were recognized as possible candidates significant in terms of clinical characteristics and selected for future investigation. Our previous research offered a more thorough elucidation of the WGCNA methodology.

### 4.12. Interaction with the Modules of Interest

The limma [50] R package provides a robust method for carrying out differential expression analysis on RNA sequencing and microarray data. The limma approach was used to discover DEGs between head and neck squamous cell carcinoma (HNSCC) and normal tissues, using the hSSC and GSE6631 datasets independently. The *p*-value was adjusted using the Benjamini–Hochberg method to control the false discovery rate (FDR). Genes exhibiting an absolute log fold change (|logFC|) of at least 1.5 and an adjusted *p*-value (adj. *p*) below 0.05 were classified as DEGs. The DEGs of the hSSC were visually represented as a volcano plot using the R programming language’s ggplot2 package. Subsequently, the genes shared between the DEGs and the co-expression genes derived from the co-expression network were used to identify potential prognostic genes. The genes were then shown as a Venn diagram using the R application VennDiagram.

### 4.13. Gene Enrichment Analysis of DEGs

The Database for Annotation, Visualization, and Integrated Discovery (DAVID; version 6.8; https://david.ncifcrf.gov/, accessed on 5 August 2022) was used to conduct a gene ontology (GO) functional analysis and a Kyoto Encyclopedia of Genes and Genomes (KEGG) pathway analysis to infer the potential functions of the DEGs. DEGs that exhibited increased expression (upregulated) and reduced expression (downregulated) were input into the DAVID online software. The ten foremost components from the cellular component (CC), biological process (BP), and molecular function (MF) categories, together with the KEGG pathways, were then organized and shown as bubble maps. The bubble plots were created using the ggplot2 R package inside the R statistical software (version 3.6.1), employing the *p*-value as the foundation for the visualization. A significance threshold of *p* < 0.05 was used to ascertain statistical significance.

### 4.14. PPI Network Construction

The protein–protein interaction (PPI) network of the differentially expressed genes associated with the hSSC aging pathway was constructed using the Search Tool for the Retrieval of Interacting Genes/Proteins (STRING; https://string-db.org/, accessed on 6 August 2022). Only interactions with a score over 0.4 were considered. The sources of active interaction were text mining, experiments, databases, co-expression, neighborhood, gene fusion, and co-occurrence. The interaction network among the DEGs and their related genes was shown, including a minimum of two interactions. The PPI network modules were analyzed using the Cytoscape software (version 3.7.1) with the Molecular Complex Detection (MCODE) plugin set to default parameters. The parameters included a degree cutoff of 2, a node score threshold of 0.2, a k-score of 2, and a maximum depth of 100. The study determined the criteria for the leading four modules, requiring MCODE scores of 2.8 or above and a minimum of three nodes. DAVID was used to perform KEGG pathway analysis on the genes inside each module. The Cytoscape plugin cytoHubba was used to choose the top 10 hub genes based on their highest degree of connectivity. The construction of the PPI network and the co-expression analysis of the hub genes were performed using STRING. The PPI network criteria were a minimum confidence score of 0.4 and a maximum of 5 interactions.

### 4.15. Validation of the Hub Genes with scRNA-Seq Datasets

Transcriptome analysis was conducted with pre-existing 10× genomic datasets (GSE149512) [51]. Detailed information on the development of these datasets may be acquired from the source documents and culture dataset. In conclusion, both datasets used human spermatogonia. The origin of the “culture” scRNA-seq dataset may be traced to spermatogonia. The transgenes serve as markers for further research and do not alter the biology of the spermatogonia population. The adult testis dataset was acquired by isolating spermatogonia by fluorescence-activated cell sorting (FACS) with the CD9Bright/ID4-eGFP+ markers, comprising over two replicates. The presence of authentic stem cells was confirmed by transplantation research. Initially, undifferentiated cultures of spermatogonia were produced using the THY1+ subset of undifferentiated spermatogonia derived from the adult testis. The cultures were then sustained for 10 weeks, or 10 passes, at conditions optimized for glycolysis (10% O_2_, 5% CO_2_). Subsequently, the spermatogonia were extracted from feeder cells and readied for single-cell RNA sequencing (scRNAseq) with three replicates. The existence of stem cells in cultures was confirmed by spermatogonial implantation. Our prior research showed that THY1+ selection efficiently isolates the ID4-GFP+ population of spermatogonia from the adult mouse testis. Despite the use of various approaches in prior research to enhance spermatogonia, the populations addressed are generally analogous.

The transcriptomes of “culture” and “testis” were imported into Seurat (version 4.03) and amalgamated into a single entity. Cells of an inferior quality or those identified as doublets were excluded from the dataset. Cells of a low quality were characterized by unique feature counts less than 200 or above 6500. Doublets were characterized as cells with above 25% mitochondrial numbers. The data were normalized using the “NormalizeData” function in Seurat; then, integration was executed employing the “Harmony” approach. The “FindVariableFeatures” program was used to identify genes with differential expression levels for use in principal component analysis. A total of 10 significant components were used for clustering and UMAP visualization, with a resolution parameter set to 0.5.

### 4.16. Analysis of Human Metastatic and Cell–Cell Interactions

The markers used for cell type identification were sourced [43]. We used the same pre-processing and classification procedures as those described for the mouse syngeneic studies. To categorize T cell subgroups, we first identified T cells from the whole dataset. We then categorized the expected T cells into separate classifications based on their attributes. CD8+ positive cells were characterized by the expression of CD8A+ and CD8B+ and the absence of CD4− markers. T-helper cells were characterized as CD8−, CD8B−, CD4+, FOXP3−, and CD25− cells. Tregs were characterized as CD8A−, CD8+, CD4+, FOXP3+, and CD25+ cells. The categorization procedure used was identical to the one previously delineated. We then calculated interaction scores and evaluated their significance using the previously outlined procedure [52].

Interaction score (receptor, ligand, cell type 1, cell type 2) =

1ncelltype1∑i∈ cell type 1ei, receptor × 1ncelltype2∑j∈ cell type 2ej, ligand

*e_i_*_;*j*_ = expression of gene *j* in cell *i*

*n_c_* = number of cells of cell type

## 5. Conclusions

In summary, after an extensive study of chromatin property data (ATAC-seq, DNase-seq, and ChIP-seq) and gene expression data (RNA-seq and microarray data), we preliminarily discovered SSC-specific transcription factors and created transcription factor-mediated gene regulatory networks during SSC formation. The principal SSC-specific transcription factors (NFIB, NFIX) and their associated SSC-specific gene targets were meticulously examined. The findings suggest that NFIB and NFIX may play a role in the control of SSC formation and spermatogenesis. Our discovery lays the groundwork for future research focused on clarifying the function of these transcription factors and their target hub genes in spermatogonial stem cell development, while also offering possible induction factors to enhance the efficiency of differentiating embryonic stem cells into spermatogonial stem cells in vitro.

## Figures and Tables

**Figure 1 ijms-25-11653-f001:**
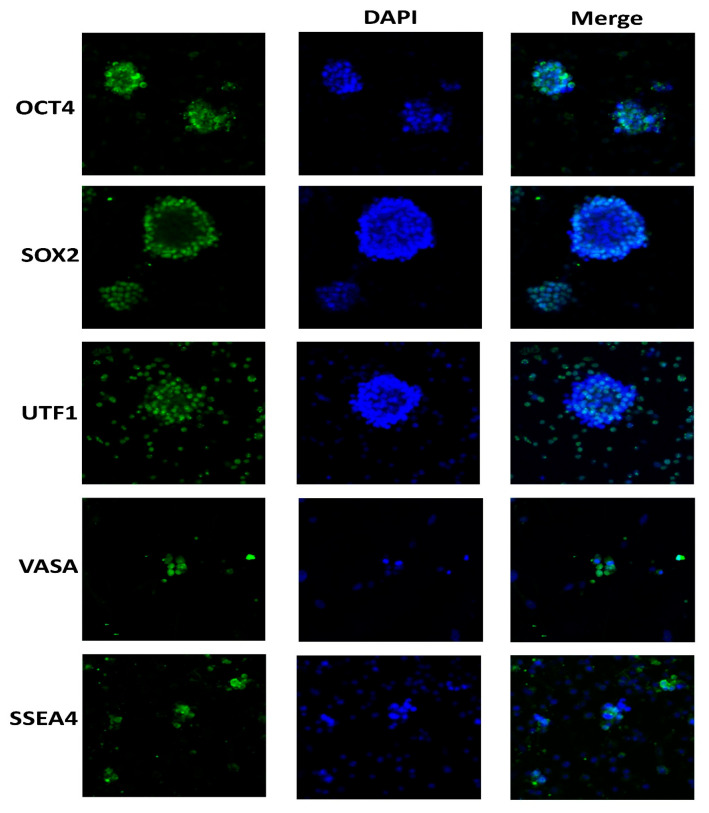
Human spermatogonia cultured in vitro after matrix and CD49f selection. The typical morphology of patient 184’s spermatogonia during culture. In every cell culture, there are connected spermatogonia in pairs, chains, small groups, or colonies. Inactivated CF1 feeder cells were used to sustain cells grown for a prolonged period of time. Human fibroblasts were absent from the cultures. OCT4, SOX2, UTF1 and SSEA4-positive somatic cells were absent from purified VASA-positive germ cell cultures.

**Figure 2 ijms-25-11653-f002:**
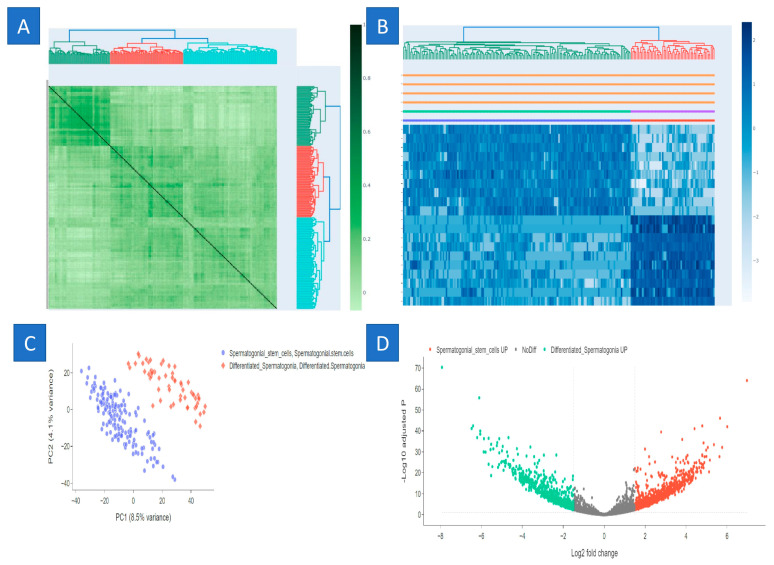
Analysis of gene expression by microarray. (**A**,**B**) Correlation plot of SSCs and fibroblasts, (**C**) volcano plot for differentially expressed genes based on microarray analysis, and (**D**) G-protein volcano plot.

**Figure 3 ijms-25-11653-f003:**
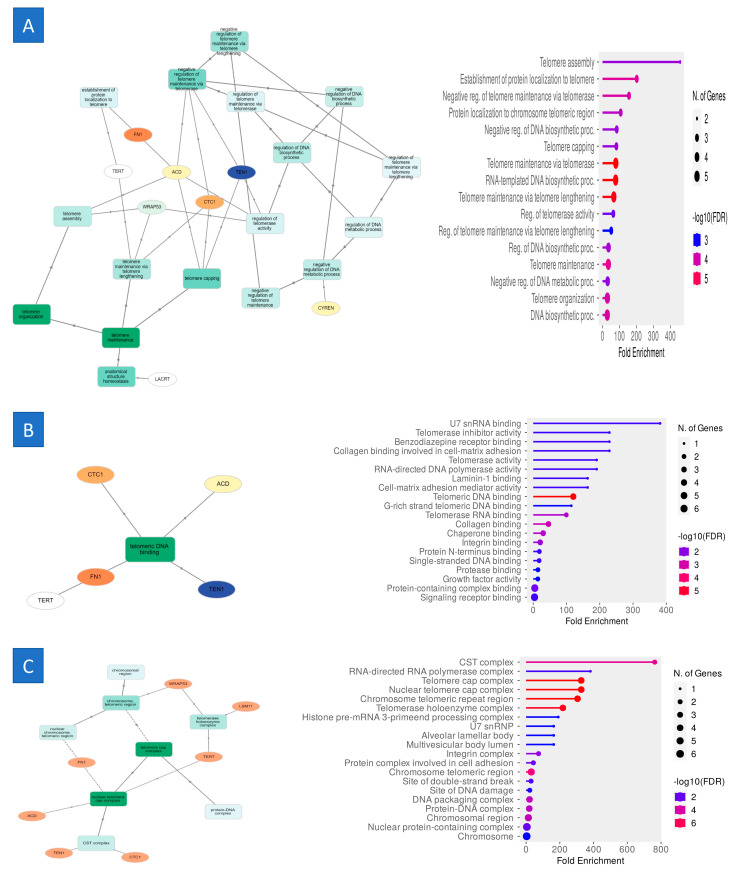
Performing gene ontology (GO) enrichment analysis on the genes inside the module. The colors correspond to the corrected *p*-values (BH), while the size of the dots corresponds to the number of genes. This picture refers to (**A**) the biological processes, (**B**) molecular functions, and (**C**) cellular components.

**Figure 4 ijms-25-11653-f004:**
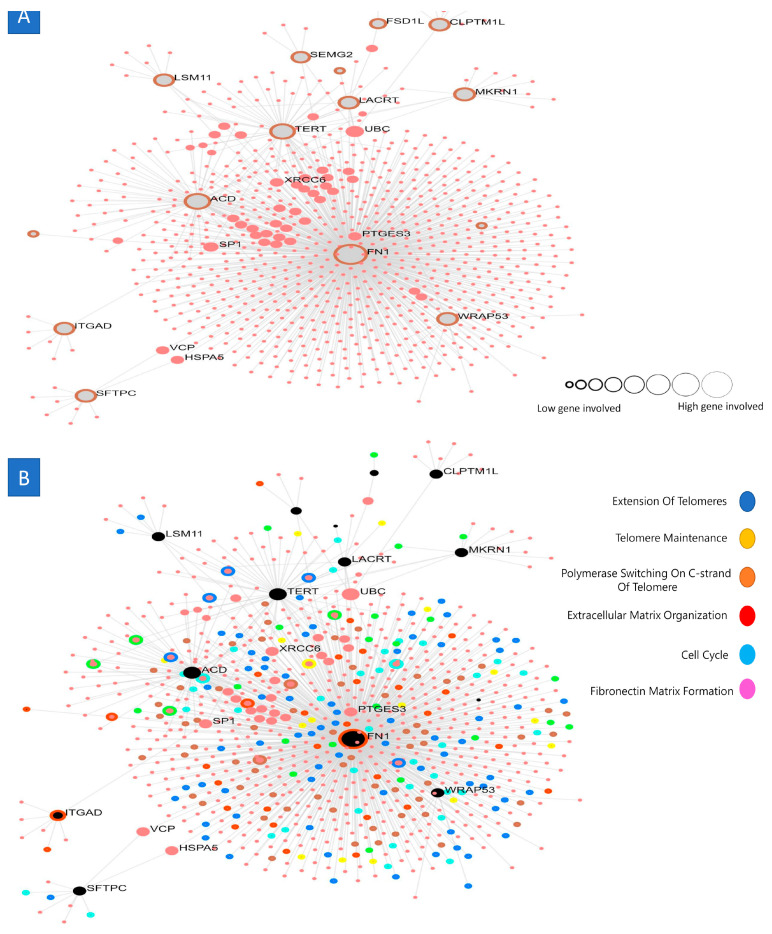
Protein–protein interaction and signaling pathway. (**A**) The protein–protein interaction (PPI) network of the core genes was established using the STRING online database. (**B**) An investigation of the signaling pathways of the core genes was conducted to ascertain their participation in signaling pathways using the KEGG database. The red nodes represent genes with a high MCC score, while the yellow nodes indicate genes with a low MCC value.

**Figure 5 ijms-25-11653-f005:**
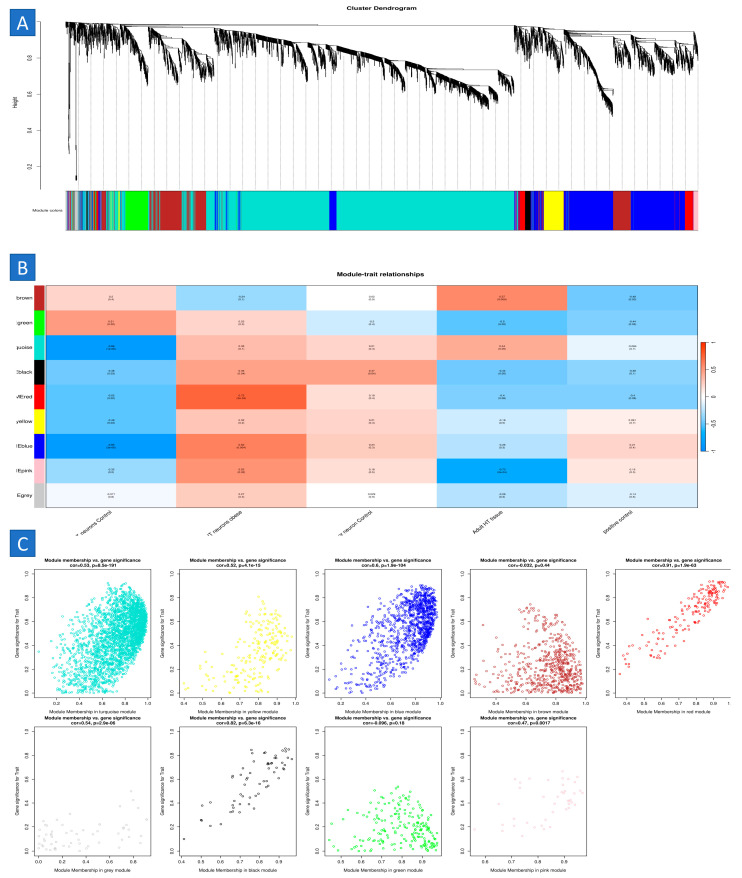
Modules linked to clinical data in SSCs are identified. (**A**) Using hierarchical gene clustering based on 1-TOM, a cluster dendrogram of co-expression network modules is created, giving each module a distinct hue. (**B**) Relationships between module and trait are shown: (**C**) each column denotes a clinical characteristic (normal or cancer), and each row denotes a color-coded module. The corresponding correlation value and related *p*-value are shown in each cell.

**Figure 6 ijms-25-11653-f006:**
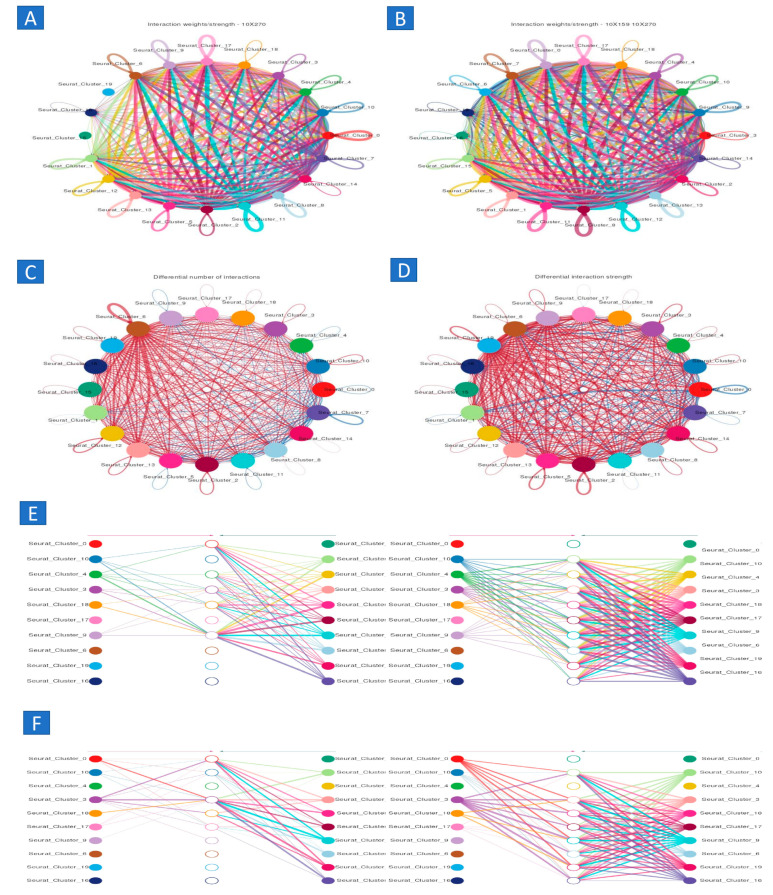
Comprehensive investigation of the single-cell transcriptome of human testis and examination of the germ cell lineage transcriptome. (**A**–**D**) The UMAP plot illustrates testicular cells derived from integrated single-cell RNA sequencing data, spanning prenatal to postnatal stages of testicular development. A total of 82,220 testicular cells are categorized into 17 color-coded clusters, representing somatic and germ cell lineages. (**E**,**F**) UMAP plots are color-coded to display the expression patterns of selected gene markers specific to somatic and germ cell lineages. Additionally, the UMAP plots categorize testicular cells by age (W, embryonic weeks; D, neonatal days; and Y, years). The focused re-clustering of 8140 germ cells identified 15 distinct clusters, each representing different stages of spermatogenesis, including undifferentiated spermatogonia (Undiff SPG), differentiating spermatogonia (Diff SPG), spermatocytes (SCytes), and spermatids (SPtids). The UMAP representation of germ cell clusters is also categorized by age, with plots color-coded to reflect the expression patterns of key markers for spermatogonia, spermatocytes, and spermatids.

**Figure 7 ijms-25-11653-f007:**
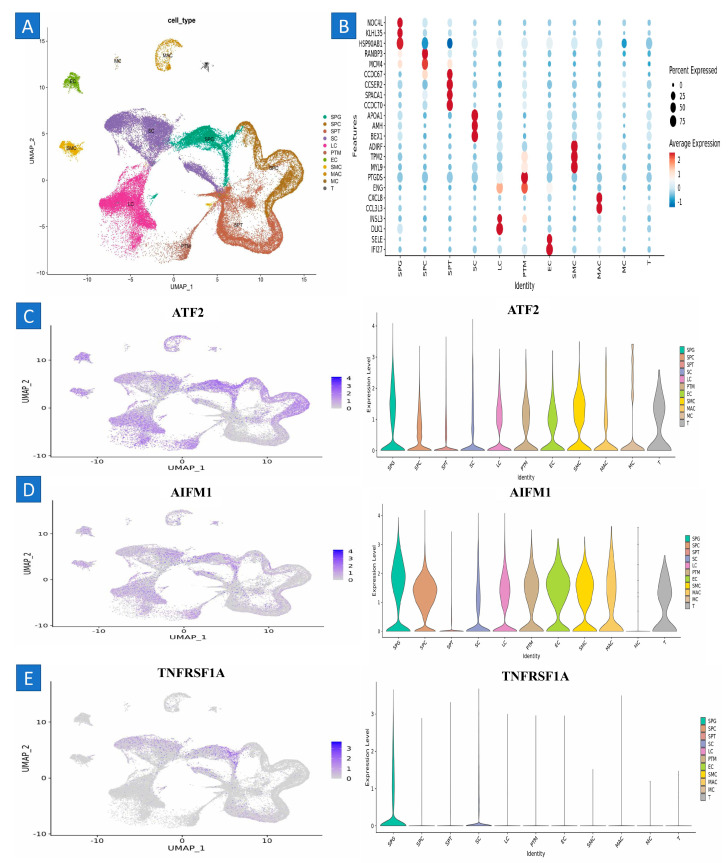
Single-cell RNA sequencing investigation of mouse pup and adult spermatogonia demonstrates similar metabolic patterns to those seen in humans: (**A**) UMAP plot depiction of germ cells from merged single-cell RNA sequencing data, (**B**) cell cluster marker, (**C**) ATF2, (**D**) AIFM1, and (**E**) TNFRSF1A. These are upregulated, suggesting their potential roles in SSCs.

**Figure 8 ijms-25-11653-f008:**
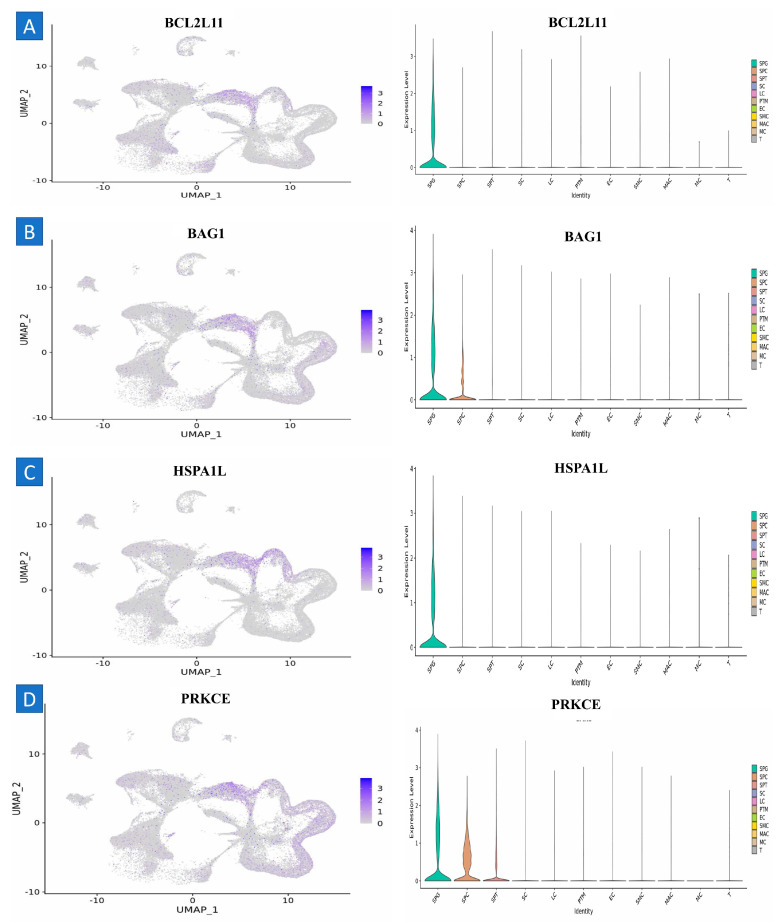
Single cell RNA sequencing analysis. (**A**) BCL2L11, (**B**) BAG1, (**C**) HSPA1L, and (**D**) PPKCE are downregulated, suggesting their potential roles in SSCs.

**Figure 9 ijms-25-11653-f009:**
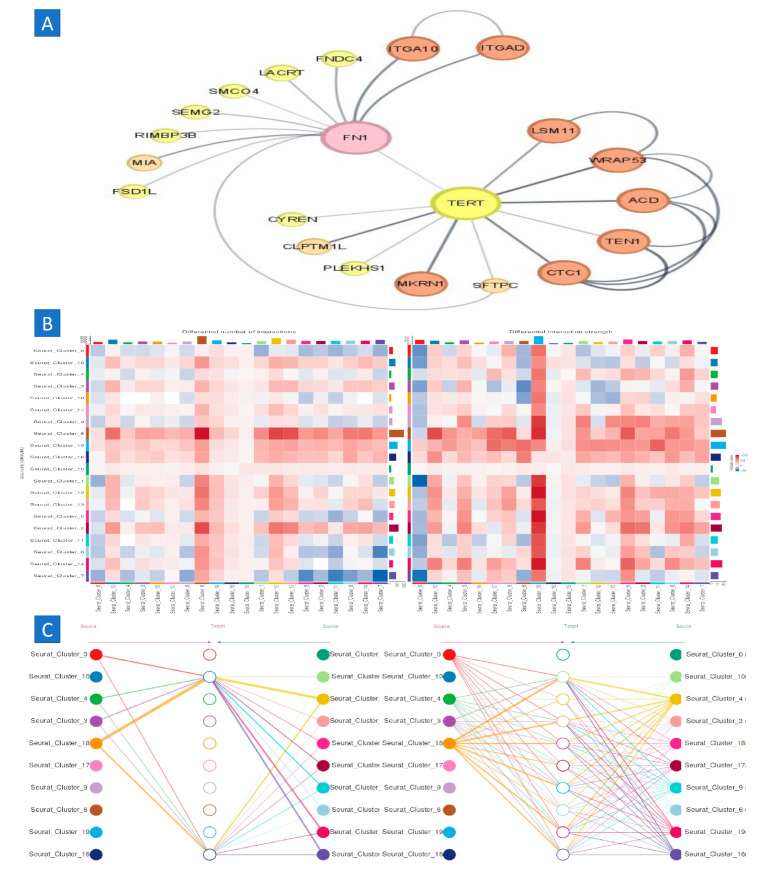
Cell–cell communication analysis. (**A**,**B**) cell–cell communicationrelated to aging, (**C**) network cell–cell communication.

**Table 1 ijms-25-11653-t001:** Aging signaling pathway gene expression changed in hSSC.

Gene Symbol	Description	Fold Change	*p*-Value
*TNFRSF10B*	tumor necrosis factor receptor superfamily, member 10b	−20.34	0.0398
*ATF2*	activating transcription factor 2	−20.04	6.25 × 10^−5^
*FAS*	Fas cell surface death receptor	−15.74	0.0001
*MAP4K5*	mitogen-activated protein kinase kinase kinase kinase 5	−14.78	5.28 × 10^−6^
*ATF7*	activating transcription factor 7	−14.36	0.0074
*TNFRSF10D*	tumor necrosis factor receptor superfamily, member 10d, decoy with truncated death domain	−10.9	0.0002
*IGF2R*	insulin-like growth factor 2 receptor	−10.6	0.045
*PRKCA*	protein kinase C, alpha	−8.95	0.007
*AIFM1*	apoptosis-inducing factor, mitochondrion-associated, 1	−8.86	0.0404
*AKT2*	v-akt murine thymoma viral oncogene homolog 2	−8.4	0.003
*HSPA8*	heat shock 70 kDa protein 8	−7.73	0.0031
*BAX*	BCL2-associated X protein	−7.46	0.0187
*MAPK14*	mitogen-activated protein kinase 14	−7.17	0.0135
*MAP4K4*	mitogen-activated protein kinase kinase kinase kinase 4	−6.43	0.0478
*JUN*	jun proto-oncogene	−6.02	0.0003
*XIAP*	X-linked inhibitor of apoptosis, E3 ubiquitin protein ligase	−5.79	0.0081
*MAP2K3*	mitogen-activated protein kinase kinase 3	−5.12	0.0002
*FOS*	FBJ murine osteosarcoma viral oncogene homolog	−3.68	0.0294
*TNFRSF1A*	tumor necrosis factor receptor superfamily, member 1A	−3.67	0.0178
*BAG3*	BCL2-associated athanogene 3	−3.55	0.0098
*MCL1*	myeloid cell leukemia 1	−3.32	0.0321
*FADD*	Fas (TNFRSF6)-associated via death domain	−3.2	0.0119
*CFLAR*	CASP8- and FADD-like apoptosis regulator	−3.05	0.017
*PRKCE*	protein kinase C, epsilon	3.08	0.0026
*HSPA2*	heat shock 70 kDa protein 2	3.32	0.0204
*BAG1*	BCL2-associated athanogene	4.1	0.003
*BCL2L11*	BCL2-like 11 (apoptosis facilitator)	5.09	0.0133
*CREM*	cAMP-responsive element modulator	7.72	0.0077
*HSPA1L*	heat shock 70 kDa protein 1-like	27.87	2.24 × 10^−5^

## Data Availability

The original contributions presented in this research are included in the article; further inquiries can be directed to the corresponding author.

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
