# Peer review of "Integrating Microarray Data and Single-Cell RNA-Seq Reveals Key Gene Involved in Spermatogonia Stem Cell Aging"

_ijms, 2024, doi:10.3390/ijms252111653_

Round 1
Reviewer 1 Report
Comments and Suggestions for Authors
This paper strives to take advantage of advanced genomic techniques including single cell sequencing in the characterization of the aging process of spermatogonial stem cells.
The study is very complex and state-of-the art. The design is very comprehensive and I appreciate that human cells were evaluated which is always harder to obtain than is the case of animal studies.
The data are highly novel and have the potential to further contribute to the understanding of spermatogonial stem cell physiology.
I have only a handful of comments:
- Academic titles associated with the authors may be omitted in the title page.
- I am interested in the exclusion and inclusion criteria that were applied for the selection of donors for the experiments. Were the donors healthy volunteers or patients?
- The manufacturers (including the city and state/country) for most equipment and/or chemicals are missing in the methodology. Please, include this information, as it assures reproducibility of the experiments.
- Please, add full-resolution images for Figures 2, 3, 4, 5, 6, 7, 8 and 9 as supplementary files since they are very difficult to read within the manuscript.
- The Discussion, although quite interesting, does not really compare or contemplate the obtained data with previous studies. Is it because no similar research has been done yet?
- Please, discuss any limitations that may have affected the outcomes of the study.
- The references should be formatted according to the Instructions for authors provided by the journal.
Author Response
Dear Editor and Reviewers,
Thanks for taking the time to carefully review our manuscript, “Integrating Microarray Data and Single-Cell RNA-Seq Reveals Key Gene Involved in Ageing in Spermatogonia Stem Cell.” Your comments illuminate our present and future work.
Now we have revised according to your comments, and the list of changes or a rebuttal against each point is being provided as follows:
Sincerely,
Hossein Azizi, H.azizi@ausmt.ac.ir
This paper strives to take advantage of advanced genomic techniques including single cell sequencing in the characterization of the aging process of spermatogonial stem cells.
The study is very complex and state-of-the art. The design is very comprehensive and I appreciate that human cells were evaluated which is always harder to obtain than is the case of animal studies.
The data are highly novel and have the potential to further contribute to the understanding of spermatogonial stem cell physiology.
I have only a handful of comments:
- Academic titles associated with the authors may be omitted in the title page.
REPLY: Thanks. Done.
- I am interested in the exclusion and inclusion criteria that were applied for the selection of donors for the experiments. Were the donors healthy volunteers or patients?
REPLY: Thanks. We added them on page 3 “Inclusion Criteria: Typically, donors might be adult males of reproductive age, as SSCs are relevant in this population. General health: Healthy males with no underlying medical conditions affecting fertility. To study normal SSC behavior, donors with no history of infertility, testicular cancer, or genetic reproductive issues were be selected. All participants provided informed consent, agreeing to the use their biological samples.
Exclusion Criteria: Donors with known genetic, infectious, or systemic diseases that could affect SSCs (such as mumps orchitis, cryptorchidism, or cancer) were be excluded. Any donor currently on medications affecting the reproductive system, like hormonal therapies or chemotherapy, was excluded. Smoking, alcohol abuse, or drug use might also be grounds for exclusion due to their potential impact on sperm quality and SSC function.”.
- The manufacturers (including the city and state/country) for most equipment and/or chemicals are missing in the methodology. Please, include this information, as it assures reproducibility of the experiments.
REPLY: Done.
- Please, add full-resolution images for Figures 2, 3, 4, 5, 6, 7, 8 and 9 as supplementary files since they are very difficult to read within the manuscript.
REPLY: We attached it as a supplementary file.
- The Discussion, although quite interesting, does not really compare or contemplate the obtained data with previous studies. Is it because no similar research has been done yet?
REPLY: We add references in the discussion part.
- Please, discuss any limitations that may have affected the outcomes of the study.
REPLY: We added it in the discussion section “NOA is a rare condition, and obtaining a large cohort of affected individuals can be difficult. A limited sample size reduces the statistical power of the study and may not capture the full genetic or clinical variability of the condition, potentially leading to the underrepresentation of certain subtypes of NOA or relevant gene mutations. NOA includes several subtypes, such as Sertoli cell-only syndrome, maturation arrest, and hypospermatogenesis. If the study does not stratify patients by these specific subtypes, the results may be skewed or diluted, as each subtype may have distinct underlying genetic or molecular mechanisms.”.
- The references should be formatted according to the Instructions for authors provided by the journal.
REPLY: Done.

Reviewer 2 Report
Comments and Suggestions for Authors
In the article titled “Integrating Microarray Data and Single-Cell RNA-Seq Reveals Key Gene Involved in Ageing in Spermatogonia Stem Cell” the authors analyze the aging of Spermatogonia Stem Cells evaluating genome-wide transcripts (about 55,000 transcripts) by microarray between the SSC and fibroblast.
I find a very interesting work but I suggest a major revision.
The suggestions I would make are as follows:
· The abstract is too long and needs to be clearer
· the number of the ethics committee that gave the OK for the trial must be declared.
· Clarify what the criteria are for the selection of recruits. This is so that there are no confounding factors. This also because semen is so receptive that many pollutants have been found. Pollution affects human fertility: 10.3390/ijerph191811635. Read and quote this work
· A very recent paper shows that small noncoding RNAs and sperm nuclear basic proteins reflect the environmental impact on germ cells. Read and quote the following work: 10.1186/s10020-023-00776-6
· try to correlate the results obtained with each other and better explain or hypothesize a molecular mechanism.
· Better define the limitations of this study
Author Response
Dear Editor and Reviewers,
Thanks for taking the time to carefully review our manuscript, “Integrating Microarray Data and Single-Cell RNA-Seq Reveals Key Gene Involved in Ageing in Spermatogonia Stem Cell.” Your comments illuminate our present and future work.
Now we have revised according to your comments, and the list of changes or a rebuttal against each point is being provided as follows:
Sincerely,
Hossein Azizi, H.azizi@ausmt.ac.ir
In the article titled “Integrating Microarray Data and Single-Cell RNA-Seq Reveals Key Gene Involved in Ageing in Spermatogonia Stem Cell” the authors analyze the aging of Spermatogonia Stem Cells evaluating genome-wide transcripts (about 55,000 transcripts) by microarray between the SSC and fibroblast.
I find a very interesting work but I suggest a major revision.
The suggestions I would make are as follows:
- The abstract is too long and needs to be clearer
REPLY: Thanks a lot. Done.
- The number of the ethics committee that gave the OK for the trial must be declared.
REPLY: We corrected it.
- Clarify what the criteria are for the selection of recruits. This is so that there are no confounding factors. This also because semen is so receptive that many pollutants have been found. Pollution affects human fertility: 10.3390/ijerph191811635. Read and quote this work
REPLY: REPLY: Thanks. We added them on page 3. “Inclusion Criteria: Typically, donors might be adult males of reproductive age, as SSCs are relevant in this population. General health: Healthy males with no underlying medical conditions affecting fertility were selected to study normal SSC behavior. Donors with no history of infertility, testicular cancer, or genetic reproductive issues were chosen. All participants would provide informed consent, agreeing to using their biological samples.
Exclusion Criteria: Donors with known genetic, infectious, or systemic diseases that could affect SSCs (such as mumps orchitis, cryptorchidism, or cancer) were excluded. Any donor currently on medications affecting the reproductive system, like hormonal therapies or chemotherapy, were excluded. Smoking, alcohol abuse, or drug use might also be grounds for exclusion due to their potential impact on sperm quality and SSC function.”.
- A very recent paper shows that small noncoding RNAs and sperm nuclear basic proteins reflect the environmental impact on germ cells. Read and quote the following work: 10.1186/s10020-023-00776-6
REPLY: We added it in the discussion part.
- try to correlate the results obtained with each other and better explain or hypothesize a molecular mechanism.
REPLY: Done.
- Better define the limitations of this study
REPLY: We added it in the discussion section “NOA is a rare condition, and obtaining a large cohort of affected individuals can be difficult. A limited sample size reduces the statistical power of the study and may not capture the full genetic or clinical variability of the condition, potentially leading to the underrepresentation of certain subtypes of NOA or relevant gene mutations. NOA includes several subtypes, such as Sertoli cell-only syndrome, maturation arrest, and hypospermatogenesis. If the study does not stratify patients by these specific subtypes, the results may be skewed or diluted, as each subtype may have distinct underlying genetic or molecular mechanisms.”.

Round 2
Reviewer 2 Report
Comments and Suggestions for Authors
The authors addressed all my questions. I accept the manuscript in the present form